# Bi-Metal Phosphide NiCoP: An Enhanced Catalyst for the Reduction of 4-Nitrophenol

**DOI:** 10.3390/nano9010112

**Published:** 2019-01-18

**Authors:** Lijie Sun, Xia Xiang, Juwei Wu, Chao Cai, Dongyi Ao, Jinling Luo, Chengxiang Tian, Xiaotao Zu

**Affiliations:** 1School of Physics, University of Electronic Science and Technology of China, Chengdu 610054, China; sunlijie09@std.uestc.edu.cn (L.S.); wu756070691@gmail.com (J.W.); aodongyi@outlook.com (D.A.); luojinling0803@gmail.com (J.L.); tianchengxiangky@163.com (C.T.); 2Institute of Fundamental and Frontier Science, University of Electronic Science and Technology of China, Chengdu 610054, China; ZChaoC@yahoo.com (C.C.); xtzu@uestc.edu.cn (X.Z.)

**Keywords:** phosphating cobalt nickel, reduction 4-NP, efficiency, hydrogenation

## Abstract

Porous phosphide Ni_x_Co_y_P composite nanomaterials are successfully synthesized at different Ni/Co ratios (=0, 0.5, 1, and 2) to reduce 4-nitrophenol. The X-ray diffraction and X-ray photoelectron spectroscopy results demonstrate that the products are CoP, NiCoP/CoP, NiCoP, and NiCoP/Ni_2_P when the Ni/Co ratio is 0, 0.5, 1, and 2, respectively. The products exhibit different catalytic performance for reduction of 4-nitrophenol at room temperature. Among them, the pure NiCoP delivers a better catalytic efficiency with $k_{\mathrm{app}}=677.4\times 10^{-2}{\;\min}^{-1}$ and $k=338.7\;\left({\mathrm{Lg}}^{-1}\min^{-1}\right)$, due to the synergy between Ni and Co atoms. The sequence of catalytic efficiency of different samples is CoP < NiCoP/CoP < NiCoP/Ni_2_P < NiCoP.

## 1. Introduction

In the past decades, water contamination has become an inevitable issue, and the use of unhealthy water is one of the most important sources of diseases [1]. Among these toxic effluents, 4-Nitrophenol (4-NP) is deemed to be one of the most ubiquitous organic contaminants in waste waters generated from agricultural and industrial production [2,3]. Therefore, many efforts are devoted to develop technologies for water purification, such as the utilization of polymeric membranes [4,5,6,7]. However, the chemical properties and toxicity of pollutants cannot be changed. And there is a broad consensus on the difficult detoxification of 4-NP by conventional water treatment due to its high chemical stability and resistance to microbial degradation which are attributed to the presence of a nitro-group in the aromatic compound [8]. Fortunately, 4-NP can be degraded to p-aminophenol (4-AP) by using the organic or inorganic catalysts [9,10], such as polymeric membranes or metal catalysts. In addition, 4-AP is also widely used as a dyeing agent, photographic developer, corrosion inhibitor in paints, and anticorrosion-lubricating agent in fuels for two-cycle engines [11]. Therefore, it is of great significance to convert the hazardous 4-NP into a useful 4-AP by hydrogenation [12,13].

Transitional-metal phosphides (such as FeP, MoP, Co_2_P, CoP, NbP and Ni_2_P) have drawn attentions recently because of their excellent semiconducting, topological Weyl semimetal, novel magnetic and photocatalytic properties [14,15,16,17,18,19,20]. Different from nitrides and carbides, the phosphorus atom in transition metal phosphides is usually found at the center of a triangular prism, rather than residing in the interstitial spaces between metal host atoms. This is because that the atomic radius of phosphorus (0.109 nm) is larger than that of nitrogen (0.065 nm) or carbon (0.071 nm) [21]. Furthermore, since phosphides have more coordination unsaturated bonds than carbide and nitride, they have more phase states than carbide and nitride.

Recent studies have demonstrated that transitional metal phosphides present better catalytic effects for hydrodesulfurization, hydrodenitrogenation and hydrogenation [22,23]. For instance, Stinner et al. fabricated the MoP with better hydrodenitrogenation performance than MoS_2_ owing to the enrichment of Mo in MoP [24]; Song et al. designed FeP/SiO_2_ as hydrodesulfurization catalysts [25]; Ibrahim et al. prepared the Ni_0.3_MoP catalysts, supported on Al_2_O_3_ as hydrodenitrogenation and hydrodesulfurization catalysts [26]. For the hydrogenation of 4-NP, phosphide materials exhibited remarkably high efficiency. Tian et al. prepared an Ni_2_P/Ni_12_P_5_ bi-phase nanocomposite, and the obtained catalysts showed better activity with $k_{\mathrm{app}}=50.04\times 10^{-2}{\;\min}^{-1}$ [8]; Wei et al. synthesized an urchin-like and hollow Ni_x_P_y_ superstructure with $k_{\mathrm{app}}=57\times 10^{-2}{\;\min}^{-1}$ [27]; Lu and his coworkers prepared Ni@Ni_x_P_y_ core-shell microstructures with $k_{\mathrm{app}}=38\times 10^{-2}{\;\min}^{-1}$ [28]; Huang et al. fabricated porous Co_2_P for hydrogenation of 4-NP with the $k_{\mathrm{app}}=0.09\times 10^{-2}{\;\min}^{-1}$ [3]; Lu et al. synthesized CoP nanoflakes with $k_{\mathrm{app}}=34\times 10^{-2}{\;\min}^{-1}$ [29].

Since both CoP and Ni_2_P, with high hydrogenation efficiency for 4-NP, can the two materials be combined together to enhance the catalytic efficiency in the premise that structure and morphology are not broken? Perhaps the synergistic effect between Ni and Co will lead to better catalytic efficiency. To our knowledge, there has no report on the catalytic performance of composite of Ni_x_Co_y_P. 

In this work, we firstly report the fabrication, structure, and performance of porous Ni_x_Co_y_P composites, and discuss the synergistic effects of Ni and Co in a hydrogenation process. The results demonstrate that the NiCoP composite nanomaterials have excellent catalytic efficiency and promising application as catalyst for 4-NP reduction.

## 2. Experimental

### 2.1. Catalyst Synthesis

The porous phosphide catalysts were prepared in three steps as the displayed in Figure 1. Firstly, NiCl_2_·6H_2_O (0 mmol, or 0.66 mmol, or 1 mmol, or 1.33 mmol), CoCl_2_·6H_2_O (2 mmol, or 1.33 mmol, or 1 mmol, or 0.66 mmol) and urea (0.2 g) were dissolved in a solution containing 20 mL deionized (DI) water and 20 mL glycol to form a transparent solution. Then, the solution was transferred into a 50-mL Teflon-lined autoclave and then kept in an oven at 140 °C for 10 h. The resulting precipitate precursors were gathered and cleaned with ethanol and DI water for several times. Secondly, the precursors were calcinated at 400 °C for 2 h in air, and black cobalt nickel oxides were obtained. Thirdly, the superfluous NaH_2_PO_2_ with the cobalt nickel oxides were mixed well and then heated to 250 °C for 2 h in Ar atmosphere. The final products were washed several times with DI water and ethanol.

### 2.2. Characterization

A powder X-ray diffraction (XRD) was conducted to determine the phase of the as-synthesized composites, with Cu Kα radiation operated at 40 kV and 30 mA. The morphologies and microstructures of the composites were characterized by employing field emission scanning electron microscope (FE-SEM, Hitachi S-4800) and Energy dispersive X-ray Spectroscopy (EDX). X-ray photoelectron spectroscopy (XPS) experiments were carried out by a Kratos XSAM 800 system with an Al Kα X-ray photoelectron spectrometer. 

### 2.3. Catalyze Measurements

The measurements for the catalytic hydrogenation of 4-NP by catalyst were conducted by UV–Vis spectroscopy in a quartz cuvette on a UV–visible spectrophotometer (TU-1901). A total amount of 3 mL of 4-NP, which was prepared by dissolving 5 mL 4-NP (1 mM) and 1 mmol NaBH_4_ reagent into 45 mL DI water was moved into a quartz cuvette. After that, different phosphide catalyst suspension was injected into the cuvette to trigger the reaction, and the UV–Vis absorbance spectra were monitored to describe the process of reaction. The reduction of 4-NP was conducted in the existence of excess freshly prepared NaBH_4_, and the reduction process can be described two steps as shown in Figure 2. Firstly, 4-NP interacts rapidly with the hydroxyl ion induced by hydrolysis of sodium borohydride. As a result, 4-NP is converted to 4-nitrophenolate ions (4-NPI). After that, the 4-NPI is reduced by NaBH_4_ and 4-AP comes is formed. All of the measurements were conducted at room temperature.

## 3. Results and Discussion

### 3.1. Characterization of Samples

Figure 3 shows the XRD patterns of the four catalysts. When the Ni/Co molar ratio is 0, 0.5, 1, and 2, the XRD patterns are typical phases of CoP, NiCoP/CoP, NiCoP, and NiCoP/Ni_2_P. Besides, the results show that NiCoP and Ni_2_P have same hexagonal structure while CoP is oethorhombic structure. For the CoP samples, the diffraction peaks at 31.58°, 36.31°, 46.16°, 48.08°, 52.13° and 56.09° can be assigned to the (011), (111), (112), (211), (103) and (020) planes of CoP (JCPDS no.65-2593), respectively. For the NiCoP samples, the diffraction peaks at 40.92°, 44.88°, 47.61°, 54.45° and 75.43° can be assigned to the (111), (201), (210), (300) and (212) planes of NiCoP (JCPDS no. 71-2336), respectively. Because the XRD patterns of Ni_2_P and NiCoP are very similar, it is difficult to distinguish the two kinds of phosphides. However, a detailed comparison of the XRD patterns of Ni_2_P and NiCoP reveals that there is a slight shift towards higher 2θ position due to introduction of Co and indicative of the formation of NiCoP ternary phosphide [30]. Meanwhile, the diffraction peaks at 40.92° in the XRD pattern of NiCoP/CoP sample could be assigned to the (111) plane of NiCoP (JCPDS no. 71-2336) and peaks at 31.58°, 36.43°, 48.10° and 52.16° could be assigned to the (011), (111), (211) and (020) planes of CoP (JCPDS no.65-2593), respectively. When the Ni/Co molar ratio is 2, the XRD pattern shows the products were the mixture of NiCoP and Ni_2_P. This indicates that the Ni/Co molar ratio plays a very important role in the phase and crystal structure of products.

To further investigate the effect of the molar ratio of Ni/Co on the products, XPS measurement of the phosphides was carried out. Figure 4a–c shows XPS spectra of Co (2p), Ni (2p) and P (2p) for the CoP, NiCoP/CoP, NiCoP and NiCoP/Ni_2_P, respectively. The Ni 2p_3/2_ peak can be fitted with three peaks at 853.1, 856.35 eV and 861.55 eV that correspond to Ni in Ni_2_P or NiCoP, Ni oxides species and the satellite peak, respectively [8,17,31,32,33]. For the CoP sample, the peaks of Co 2p_3/2_ located at 778.2 and 782.0 eV correspond to the Co-P bond and Co oxides, respectively [29,34,35]. For the other three samples, the peaks located at 778.50 eV for Co are ascribed to the Co-P bonds in NiCoP. The blue shift (0.22 eV) of Co 2p_3/2_ peak shows unambiguously that Co has a strong coupling effect and change the surface electronic landscape [36,37]. Besides, the binding energies of P 2p region show a presence of two peaks at 129.1 and 133.2 eV belonged to metal phosphide bonds and P oxides, indicating that P is not involved in the process of adjusting multi-coupling synergetic effects in all samples [14,34,38]. The observation of oxide species of Co, Ni and P in the XPS results is due to the spontaneous oxidation of samples exposed to air [39]. All these results demonstrate that the products were CoP, NiCoP/CoP, NiCoP and NiCoP/Ni_2_P when the Ni/Co ratios were 0, 0.5, 1, and 2, respectively.

Recent studies have shown that both NiCoP and Ni_2_P consist of interlinked trigonal prisms centered by P atoms [40]. This structure of NiCoP can be described as P atoms-sharing stacks. In this work, it is obvious that the CoP can be produced when the Ni/Co = 0. When the molar ratio of Ni equals to or higher than Co, NiCoP phase appears in the XRD patterns. Besides, the NiCoP and Ni_2_P show the same hexagonal structure in this work. It is reasonable to make the assumption that Co atoms partially substitute Ni atoms of Ni_2_P to form NiCoP phase, which is similar with the NiMoP system reported in the literature [41]. However, when the molar ratio of Ni is lower than that of Co, a mixture composition of NiCoP/CoP was produced. This result shows that not all of Ni atoms in the Ni_2_P phase can be replaced by Co atoms. Therefore, the Ni/Co ratio has a significant influence on the formation of products.

The effect of a different Ni/Co molar ratio on the microstructure of products were explored by SEM. Figure 5 shows the SEM images of the precursors and corresponding oxides prepared with different Ni/Co molar ratios. The images indicate that the sample consists of nanorods when the molar ratio of Ni is lower than that of Co and consists of the mixture of nanoflakes and nanorods when the molar ratio of Ni equals to or higher than Co. This implies that it tends to form nanoflakes at larger Ni/Co molar ratio.

Figure 6 demonstrates that the phosphide samples with different Ni/Co molar ratios are composed of porous nanostructures. The formation of this specific morphology and structure is attributed to the gas release during the calcination. Furthermore, the EDX-Mapping images indicate the homogeneous distribution of P, Co and Ni elements on the Ni_x_Co_y_P, as shown in Figure 7. The uniform distribution of each element reveals the successful synthesis of Ni_x_Co_y_P nanostructure.

### 3.2. Catalytic Activity

Previous reports have testified that the reduction reaction of 4-NP is difficult without catalysts. Nevertheless, the 4-NP can be rapidly reduced when catalysts involve in the reaction. A UV–Vis spectroscopy is employed to monitor the reduction process in this work. As shown in Figure 8a, the maximum absorption at 400 nm in the UV–Vis spectra reduces rapidly because of the reduction of 4-NP. At the same time, another peak emerges at 300 nm owing to the formation of 4-AP [42]. A few minutes later, the absorption peak of 4-NP disappears and color of the solution fades from dark yellow to transparent. Therefore, it is reasonable to monitor the intensity of the absorption peak at 400 nm to investigate the kinetic process of the reduction reaction. For comparison, the catalytic activity of CoP, NiCoP/CoP, NiCoP and NiCoP/Ni_2_P were tested under the same condition, and the amount of each catalyst is 0.06 mg. In Figure 8a, the decrease of peak at 300 nm is owing to the precipitation of catalysts in quartz cuvette during optical absorption measurements. These results are consistent with literatures [8,43]. Figure 8b displays the conversion-time dependent curves of 4-NP to 4-AP for various catalysts. 93% conversion of 4-NP is selected for comparison and the conversion time was 9 min, 6 min, 3 min and 4 min for the different catalyst, respectively. The catalytic activity of the four catalysts follows the sequence of NiCoP/Ni_2_P>NiCoP>NiCoP/CoP>CoP. The results demonstrate that the enhancement of catalytic activity of NiCoP sample. There is an induction period in catalytic reaction and it is an initial slow stage of a chemical reaction. After the induction period, the reaction accelerates [44]. As shown in Figure 8b, the induction period was 2 min, 1 min and 1 min for CoP, NiCoP/CoP and NiCoP/Ni_2_P, respectively. However, the induction period vanishes and a faster reaction rate was observed for NiCoP.

The hydrogenation process of 4-NP follows the pseudo-first order kinetics, which can be described with a kinetic equation:$$\ln(C_{t}/C_{0})=\ln(A_{t}/A_{0})=-k_{app}t,$$
where $A_{t}$ is the absorbance at 400 nm at time *t*; $A_{0}$ is the initial absorbance of the reactants; *t* is the reaction time; $k_{app}$ is the apparent rate constant (s^−1^), respectively. Thus, the apparent rate constant ($k_{app}$), which is determined by the spectrophotometric data, can give information on the reaction rate [45]. Figure 9a illustrates the relationship of $\ln(A_{t}/A_{0})$ versus reaction time plots in presence of 0.06 mg catalysts. The observed $k_{\mathrm{app}}$ for NiCoP, NiCoP/Ni_2_P, NiCoP/CoP and CoP is $677.4\times 10^{-2}{\;\min}^{-1}$, $\;312.6\times 10^{-2}{\;\min}^{-1}$, $259.8\times 10^{-2}{\;\min}^{-1}$ and $163.8\times 10^{-2}{\;\min}^{-1}$, respectively. In addition, the effect of different catalyst dosages for 4-NP reduction is also investigated. Therefore, the parametric k is measured, which is related to the catalyst amount, and is defined as [46]:$$k=\frac{k_{app}}{m/V},\;$$
where *m* is the mass of catalysts, and *V* is the volume of the catalytic system. In Table 1, the k values were calculated. The observed k for NiCoP, NiCoP/Ni_2_P NiCoP/CoP and CoP is $338.7\;\left({\mathrm{Lg}}^{-1}\min^{-1}\right)$, $156.3\;\left({\mathrm{Lg}}^{-1}\min^{-1}\right)$, $129.9\;\left({\mathrm{Lg}}^{-1}\min^{-1}\right)$ and $81.9\;\left({\mathrm{Lg}}^{-1}\min^{-1}\right)$, respectively.

Figure 9b shows that the catalytic efficiency varies with the Ni/Co molar ratio and indicates that NiCoP is a superior catalyst for reduction the 4-NP. Furthermore, both NiCoP/CoP and NiCoP/Ni_2_P have lower efficiency than NiCoP, which indicates the higher hydrogenation activity of the ternary phosphide (NiCoP) than the binary phosphide (CoP or Ni_2_P). Compared with other reported phosphide catalysts as shown in Table 1, NiCoP is with better catalytic efficiency. In addition, as shown in Table 1, the activity of NiCoP is even better than some noble metal catalysts and rGO catalysts, which further exhibits the superior catalytic ability of the ternary phosphides for 4-NP reduction. NiCoP was fabricated by the process of Co atoms partially substituting Ni atoms of Ni_2_P, so the higher catalytic activity of NiCoP than Ni_2_P can be ascribed to the increased Co sites in the NiCoP sample [21]. NiCoP has Ni^α+^ (0 ≤ α ≤ 1) and Co^β+^ (0 ≤ β ≤ 1) sites, indicating that Co has Co^2+^, Co^+^ and Co^0^ states [22]. As proved by Lining Ding, β will be very close to zero when the number of Ni atoms is more than that of Co atoms [21]. Thus, the increase of Ni atoms leads to the reduction of catalytic performance and the interaction of electron was modified by the existence of neighboring Ni. So its superior catalytic ability of NiCoP can be attributed to the synergy between Ni and Co and the increased electron density of Co sites. The mechanism of 4-NP reduction can be described as follows: Firstly, P acts as the proton-acceptor and combines with H which generates from the dissociation of B-H and Co combines with ${\mathrm{BH}}_{3}^{-}$. Secondly, $e^{-}$ captured from ${\mathrm{BH}}_{3}^{-}$ transmits to P under the help of the captured $e^{-}$, and the P receives a H from the O-H. Thirdly, H is delivered from P to Co at the same time, the H at the Co surface reacts with 4-NP to yield 4-AP. Finally, the 4-AP is desorbed from the surface of NiCoP [51].

In addition, the effect of different catalyst dosage on the reaction has also been observed in Figure 9c. The results show that the $k_{app}$ increases linearly with the increasing amount of the catalyst, due to the increase of reaction sites.

## 4. Conclusions

A series of phosphide catalysts, CoP, NiCoP/CoP, NiCoP and NiCoP have been fabricated. The Ni/Co ratio has important influence on the formation of phosphide and the activity for 4-NP decomposition. Among the phosphides, NiCoP catalyst showed the best performance of 4-NP decomposition due to the synergy between Ni and Co atoms. Compared with Ni^α+^, Co^β+^ exhibits a higher efficiency of reduction 4-NP because of the electronic interactions among Co, P and Ni atoms. So the increased electron density of Co sites explains the enhanced activity of the NiCoP catalyst.

## Figures and Tables

**Figure 1 nanomaterials-09-00112-f001:**
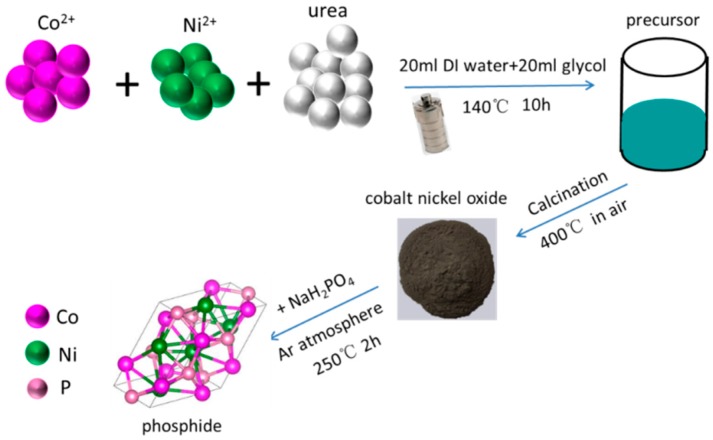
Schematic depiction of the synthesis process.

**Figure 2 nanomaterials-09-00112-f002:**
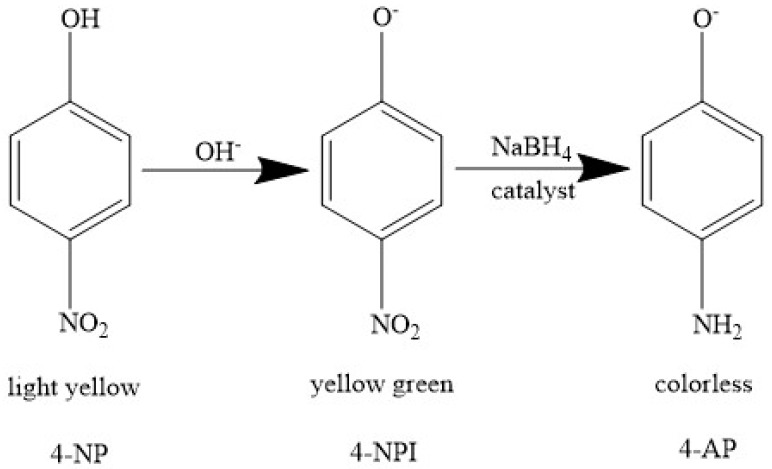
Conversion from 4-NP to 4-AP.

**Figure 3 nanomaterials-09-00112-f003:**
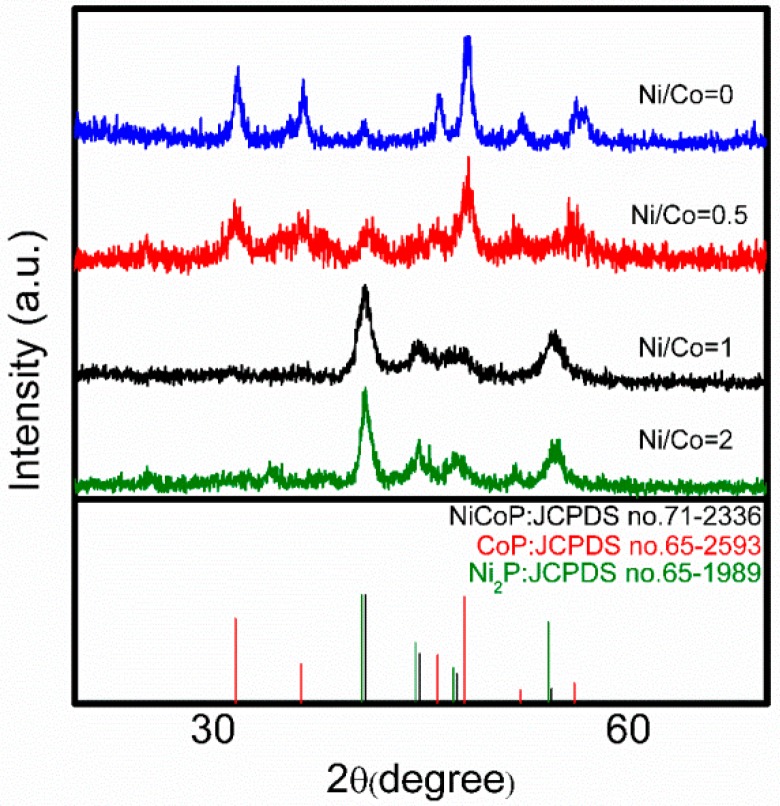
XRD patterns of the Ni_x_Co_y_P which prepared with different Ni/Co molar ratios.

**Figure 4 nanomaterials-09-00112-f004:**
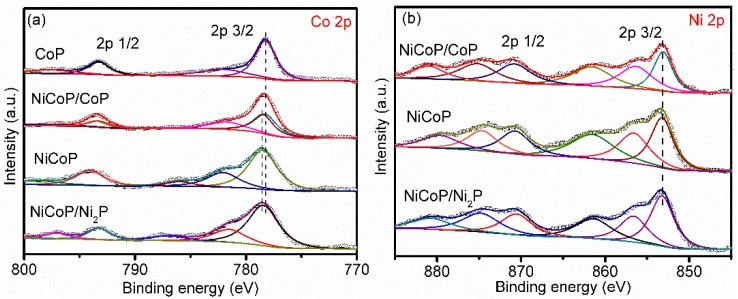
The high-resolution Co 2p (**a**), Ni 2p (**b**) and P 2p (**c**) XPS spectra of CoP, NiCoP/CoP, NiCoP and NiCoP, respectively.

**Figure 5 nanomaterials-09-00112-f005:**
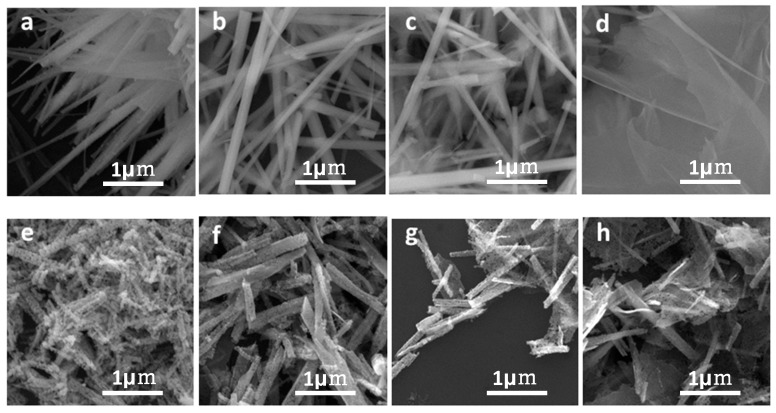
SEM images of (**a**–**d**) different precursors prepared by different Ni/Co molar ratios, and (**e**–**h**) the corresponding oxides.

**Figure 6 nanomaterials-09-00112-f006:**
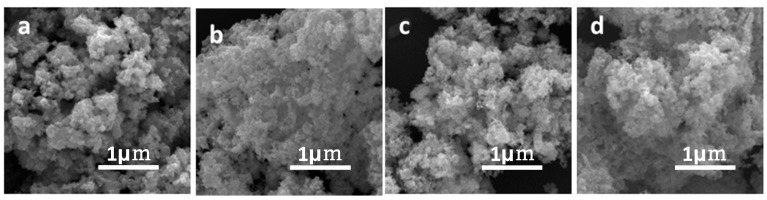
SEM images of (**a**) CoP, (**b**) NiCoP/CoP, (**c**) NiCoP and (**d**) NiCoP/Ni_2_P.

**Figure 7 nanomaterials-09-00112-f007:**
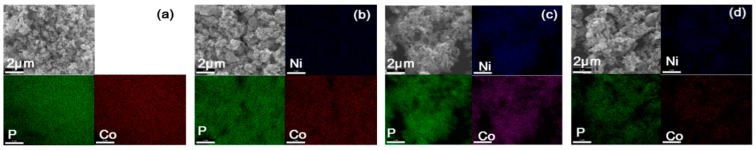
SEM images and corresponding to SEM-EDS mapping images reflects P, Co and Ni atom distributions for the (**a**) CoP, (**b**) NiCoP/CoP, (**c**) NiCoP and (**d**) NiCoP/Ni_2_P.

**Figure 8 nanomaterials-09-00112-f008:**
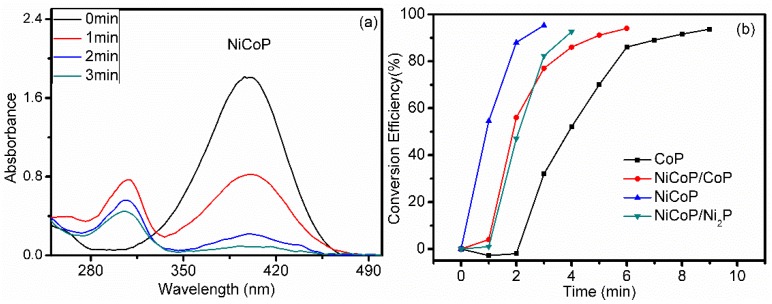
(**a**) UV–Vis spectra of 4-NP catalyzed in presence of NiCoP at different time. (**b**) The relationships between reduction efficiency of 4-NP and reaction time in the present of CoP, NiCoP/CoP, NiCoP and NiCoP/Ni_2_P, respectively.

**Figure 9 nanomaterials-09-00112-f009:**
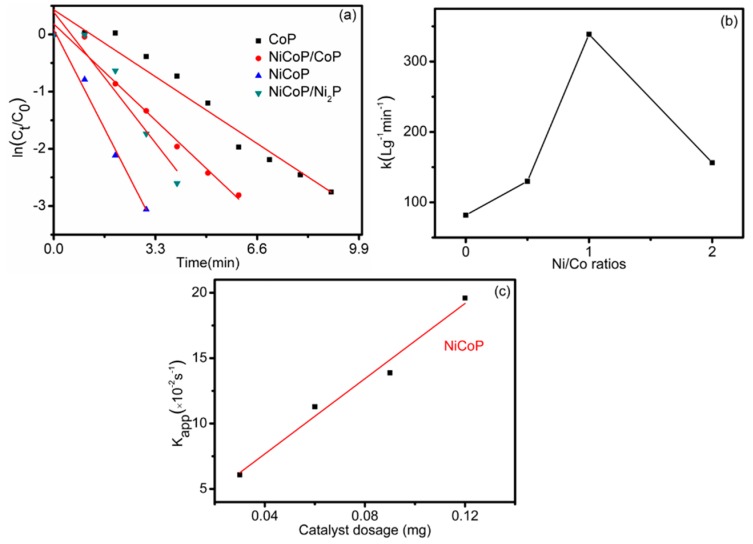
(**a**) The plots of ln(C_t_/C_0_) versus the reaction time for the reduction of 4-NP catalyzed by CoP, NiCoP/CoP, NiCoP, and NiCoP. (**b**) The catalytic efficiency varies with the Ni/Co molar ratio. (**c**) Effect of catalyst dosage on reduction of 4-NP catalyzed by NiCoP catalysts in the presence of NaBH_4_.

**Table 1 nanomaterials-09-00112-t001:** Comparison of the catalytic $k_{app}$ and k in this work with other phosphide catalysts previously reported for the reduction of 4-NP.

Catalyst	Ni/Co Ratios	$k_{app}\;$ $\left(\times 10^{-2}\;\min^{-1}\right)$	k $\left({\mathrm{Lg}}^{-1}\min^{-1}\right)$	
CoP	0	163.8	81.9	This work
NiCoP/CoP	0.5	259.8	129.9	This work
NiCoP	1	677.4	338.7	This work
NiCoP/Ni_2_P	2	312.6	156.3	This work
Co_2_P	-	0.09	0.45	[3]
CoP	-	34	34	[29]
Ni_2_P/Ni_12_P_5_	-	50.04	1.0	[8]
Ni@Ni_x_P_y_	-	38	1.58	[28]
Ni_x_P_y_	-	57	8.54	[27]
FeCo	-	49.68	24.84	[47]
Co_0.85_Se-Fe_3_O_4_	-	-	1.18	[11]
AuPd	-	-	106.56	[48]
AgPdNCs/rGO	-	3.65	98.55	[49]
AgNPs/SiNSs	-	481.14	24.06	[50]
Ni/rGO@Au	-	52.38	14.90	[13]

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
