# Peer review of "Bi-Metal Phosphide NiCoP: An Enhanced Catalyst for the Reduction of 4-Nitrophenol"

_nanomaterials, 2019, doi:10.3390/nano9010112_

Reviewer 1 Report

The authors have well revised their paper. However, several errors are still present in the bibliography section and this should be addressed in the proofs stage:

1)      Refs. 1, 5, 6, 7, 9, 16, 18, 19, 20, 21, 24, 25, 26, 30, 31, 37, 38, 39, 40, 41, 43, 44, 46, 50, 51: missing abbreviation of the journal name

2)      Ref. 4: year is 2019, “Gianluca DP” should be “Di Profio G”

3)      Ref 12: journal is missing

4)      Ref. 33: “Larcipretev” should be “Larciprete”

5)      Ref. 34: the abbreviation of the journal name should be “J. Am. Chem. Soc.”

Reviewer 2 Report

I am satisfied by the changes made by the authors

Reviewer 3 Report

The authors have incorporated all reviewers comments/suggestion. This manuscript could be accepted in its current format.

Reviewer 4 Report

No further comments

Reviewer 5 Report

My comments were well answered. The revised manuscript can be published.

This manuscript is a resubmission of an earlier submission. The following is a list of the peer review reports and author responses from that submission.

Round  1

Reviewer 1 Report

 The paper by Sun et al. reports that bi-metal phosphide NiCoP could be used as a catalyst for the reduction of 4-nitrophenol. The degree of innovation is very good. The potential impact is good. However, some issues prevent me from accepting the current version of the manuscript for publication.

1)      The authors should report in the Introduction that the use of unhealthy water is one of the most important sources of diseases, especially in developing countries. Therefore, many efforts are devoted to develop technologies enabling water purification through the removal of contaminants, in most cases using polymeric membranes [1-4].

2)      In the list of transition-metal phosphides, NbP should be inserted too due to its properties as topological Weyl semimetal, with a couple of references [5, 6].

3)      In the sentence, “The Ni 2p3/2 121 peak can be fitted with three peaks at 853.1, 856.35 eV and 861.55 eV that correspond to Ni in 122 Ni2P or NiCoP, Ni oxides species and the satellite peak, respectively [3, 10, 22, 23]”, also the XPS study on water/Ni(111) in Ref. [7] should be mentioned for completeness.

4) In XPS, usually the binding energy (BE) is reported in the x-axis with reverse scale, i.e. higher BE on the left.

[1]       Overcoming temperature polarization in membrane distillation by thermoplasmonic effects activated by Ag nanofillers in polymeric membranes, Desalination (2018) doi:10.1016/j.desal.2018.03.006.

[2]       Photothermal membrane distillation for seawater desalination, Adv. Mater. 29 (2017) 1603504.

[3]       The advent of graphene and other two-dimensional materials in membrane science and technology, Curr. Opin. Chem. Eng. 16 (2017) 78.

[4]       When plasmonics meets membrane technology, J. Phys.: Condens. Matter 28 (2016) 363003.

[5]       Chiral magnetoresistance in the Weyl semimetal NbP, Sci. Rep. 7 (2017) 43394.

[6]       Resistivity of Weyl semimetals NbP and TaP under pressure, physica status solidi (RRL) - Rapid Research Letters 11 (2017) 1700182.

[7]       Unveiling the Mechanisms Leading to H2 Production Promoted by Water Decomposition on Epitaxial Graphene at Room Temperature, ACS Nano 10 (2016) 4543.

Reviewer 2 Report

Interesting paper about porous phosphide composite nanomaterials, capable to catalyze the reduction of 4-nitrophenol at room temperature.  The paper is well written and presented, it reports novel results of sufficient significance, and could be published in nanomaterials provided the authors address the following minor comments:

a) UV/Vis spectra in Fig. 8 are redundant: rather, points extracted at a single wavelength vs time will be more self-explanatory on the reaction kinetics;

b) please cite relevant references on organc nanostructured materials and catalysts: 10.1021/acs.chemrev.8b00286; doi: 10.1021/jacs.7b03412; doi:10.1021/acs.joc.6b01922.

Reviewer 3 Report

The manuscript entitled “Bi-metal phosphide NiCoP: an enhanced catalyst for the reduction of 4-nitrophenol” has described the synthesis of porous phosphide NixCoyP composite nanomaterials at different Ni/Co ratios (=0, 0.5, 1, and 2). Further, the synthesized catalyst has been utilized for the water treatment as 4-nitrophenol reduction. The present work is good and will be beneficial for the researcher working in this field. The manuscript could be accepted in esteemed journal “Nanomaterials” after incorporating following comments/suggestions.

Please check the manuscript for typographical error.

Some relevant references are missing. The following references should be cited and discussed in the main body of the manuscript (Nanoscale Res Lett. 2017, 12: 7, DOI:10.1186/s11671-016-1776-z; ChemCatChem 2016, 8, 690-693; Green Chemistry 2016, 18, 1327-1331; Green Chemistry 2016, 18, 1019-1022).

For better understanding of the morphology authors should provide TEM analysis of the synthesized catalyst.

Reviewer 4 Report

Dear authors,
As you already pointed in your manuscript the reduction of 4-NP into 4-AP is not a new topic. I think that the manuscript has not really captured advances in the state of the art (in the context of nanomaterials).
The use of transition metals for his purpose is already described in detail in works such as:
Application of NiCo2O4as a catalyst in the conversion of p-nitrophenol to p-aminophenol
Volume 62, Issue 23, 31 August 2008, Pages 3900-3902. Materials Letters; where the good performance of NiCoP catalyst for 4-NP decomposition is already reported.
It will however interesting to compare this somehow “more classical approach” with other approximations using emerging nanomaterials like Graphene Oxide.
Just as an additional comment you have contextualized the interest of the 4-NP reduction into 4-AP targeting water decontamination (many other authors also contextualize the interest of this 4-NP degradation in the same way). The degradation into 4-AP, despite the potential lesser toxicity, does not fully solve the problem of water contamination. This method does not constitute an actual absorption or removal of the reagents.
Overall your contribution fits better in a journal like MDPI-Catalysts rather than in MDPI-Nanomaterials.

Reviewer 5 Report

The manuscript described the synthesis and characterization of NiCoP composite materials which were used as catalyst for the reduction of 4-nitrophenol in aqueous solution. The presented results show that the phase composition of the synthesized materials strongly depends on the applied molar Co to Ni ratio used for synthesis of the cobalt nickel oxide. STEM image and results of mapping show that the single elements are well distributed inside the final composite material. Furthermore, it was demonstrated that the composite materials differ in their catalytic activity for 4-nitrophenol reduction. Differences in catalytic activity were attributed to variation in phase composition. I recommend publishing the manuscript after minor revision.

Remarks

The XRD pattern showed only the formation of P containing Co, Ni or bimetallic Co/Ni phases. In XPS, oxidic species were detected for Co, Ni and P. It has to be explained why the oxidic species can be found in XPS and what that means. Contains the composite materials also oxidic phases?

Figure 8 shows the temporal evolution of the UV/Vis spectra for the different catalysts. The UV/Vis spectra show a continuous decrease of UV/Vis light absorbance for the peak at 400 nm which was attributed to the transformation of 4-nitrophenol. With the decrease of UV/Vis absorbance of the peak at 400 nm a new peak at 300 nm appears. This new peak was assigned to the formation of p-aminophenol (4-AP). Why does the increase in absorbance for the peak at 300 nm (4-AP) does not correlate with the decrease in absorbance of the peak at 400 nm (4-NP)? Are there formed other products than 4-AP?

In general, the manuscript might be more suitable for Catalysts than for Nanomaterials.
